# Impact of Rainfall Movement on Flash Flood Response: A Synthetic Study of a Semi-Arid Mountainous Catchment

**Shahin Khosh Bin Ghomash** [1,2,*], **Daniel Bachmann** [1], **Daniel Caviedes-Voullième** [3,4,5] **and Christoph Hinz** [2]

1 Research Group Flood Risk Management, Magdeburg-Stendal University of Applied Sciences, 39114 Magdeburg, Germany; daniel.bachmann@h2.de

2 Chair of Hydrology, Brandenburg University of Technology Cottbus-Senftenberg, 03046 Cottbus, Germany; christoph.hinz@b-tu.de

3 SimDataLab Terrestrial Systems, Jülich Supercomputing Centre, Forschungszentrum Jülich, 52428 Jülich, Germany; d.caviedes.voullieme@fz-juelich.de

4 Institute for Bio-Geosciences: Agrosphere IBG-3, Forschungszentrum Jülich, 52428 Jülich, Germany

5 Centre for High Performance Scientific Computing Terrestrial Systems (HPSC-TerrSys), Geoverbund ABC/J, 52428 Julich, Germany

* Correspondence: shahin.khoshbinghomash@h2.de

**Abstract:** Rainfall is a spatiotemporally varied process and key to accurately capturing catchment runoff and determining flood response. Flash flood response of a catchment can be strongly governed by a rainfall's spatiotemporal variability and is influenced by storm movement which drives a continuous spatiotemporal change throughout a rainfall event. In this work, the sensitivity of runoff and flooded areas to rainfall movement are assessed in the Kan catchment (Iran). The allochthonous nature of floods in the catchment and how they interact with the effects of rainfall movement are investigated. Fifty synthetic rain hyetographs are generated and traversed over the catchment under different velocities and directions and used to force a 1D/2D hydrodynamic model. The results suggest rainfall movement affects the runoff response in different degrees. Peak discharge, hydrograph shapes and flooded areas are affected. Storms with higher velocities result in higher peaks and faster onsets of runoff and consequently higher flooded areas in comparison to slower storms. The direction of the movement also plays a role. Storms moving along the average direction of the stream result in higher peaks and flooded areas. The relevance of storm direction is greater for slow moving storms. Additionally, the influence of rainfall movement is modulated by hyetograph structure, and the allochthonous behavior is greatly dependent on the location within the drainage network at which it is assessed.

**Keywords:** storm movement; rainfall/runoff simulation; runoff generation; flash-flood; rainfall variability

## 1. Introduction

Rainfall is a complex, spatial and temporally varied process [1] and one of the core inputs for hydrological and hydrodynamic modeling. While rainfall movement can be the cause of continuous change in observed rainfall signals, most rainfalls are known to be moving storms with the directions varying in different seasons [2–4]. Rainfall movement is known to be an important influence on runoff generation, affecting both peak discharge and hydrograph structures [5–8]. Therefore, exploring to what extent rainfall dynamics may affect runoff generation and flooding can be an asset for adequate river discharge, flood extent estimation, and finally flood risk analysis.

Flash floods are among the most destructive natural hazards in the world with severe socioeconomic and environmental impacts on affected areas every year [9,10]. Flash floods can be the result of extreme rainfall events with high intensities, characterized by rapidly rising, fast-moving flows which in turn can result in devastating damages [11]. Flash floods are also known to have the highest average mortality rates in comparison to other flood

types [12]. Multiple factors such as terrain gradient, soil type, vegetation cover, as well as precipitation can influence the occurrence of flash floods [10,11,13]. In steep, rocky, or sealed landscapes, or within heavily urbanized areas, flash floods can occur even as a result of relatively small amounts of rain [11,14]. The mentioned areas typically consist of complex geometries which promote trans-critical and complex flow patterns [15] and can therefore be demanding in terms of hydrodynamic modeling. Currently operational forecasting systems for flash floods based on hydrodynamic models have been developed for such areas [16,17]. Nonetheless, numerical modeling offers an effective means to understand the dynamic and complex process of flash floods and how they might interact with other processes such as precipitation [13,18,19].

In some catchments with relatively large areas, rainfall might take place only in the mountainous areas, while the flooding often manifests through the rivers in the plains further downstream, in which little or no rainfall is observed during the flash-flood event. This form of flash-flood may be termed as allochthonous river flooding, as the origin of the water is in a different place than the flood. The allochthonous nature of rivers has been subject to some studies in the past. For example, Deodhar et al. [20] found a decrease in the width–depth ratio of rivers in the Deccan Trap region (India) due to the allochthonous nature of the rivers and the consequence changes in discharge. Wrzesiński et. [21] studied the changes in the flow regime of the Vistula River in Poland through the years 1971–2010 and concluded that the allochthonous nature of the river plays a significant role in the alternations of the river´s flow regime. While many floods such as the 2013 flooding of the Elbe river in Magdeburg, Germany may be classified as allochthonous [22], it is worthwhile studying how different processes such as rainfall and their dynamics may interact with such floods. Furthermore, these types of floods can be mainly classified as river floods, which have seen an increase in recent years throughout the world [23]. This merits the question: how do the allochthonous nature of a river, runoff scale effects and rainfall movement interact with flood dynamics?

Previous studies have focused on how storm movement affects runoff generation both in the form of laboratory experiments [24,25] or by means of numerical approaches [7,26–29]. Amengual et al. [30] studied the role of storm movement in the flash flood response of a 2458.3 $km^2$ catchment in Spain to a real storm event from 2012. They found that storm movement caused an earlier time of peak discharge and increased the peak discharge throughout that event. Perez et al. [31] studied the effects of storm direction on peak flow frequency analysis in two watersheds in the U.S.A. They used stochastically generated rain fields as input for a lumped hydrological model and showed that peak flow distributions are highly impacted by storm direction. Sigaroodi et al. [32] studied how the consideration of storm movement may affect rainfall/runoff modeling results in a 1146 $km^2$ catchment in Iran. Seven storms were identified by analyzing rain gauge records of the area and shifted based on estimated velocities and directions. The study revealed that using shifted hyetographs, which consider the rainfall movement over sub-basins, decreased differences between simulated and observed hydrographs for the considered events. Nikolopoulos et al. [29] studied the effect of storm movement on the flood shape of the 700 $km^2$ area Fella River basin in the eastern Italian Alps in response to a real storm event from 2003. They concluded that storm motion has an almost negligible impact on the flood response of the catchment. Seo et al. [7] studied the influence of storm movement on peak discharge using a conceptual model based on characteristic timescales and showed that the peak response behavior of moving rainfalls varies from those of stationary rains due to the interdependence between rainfall duration and storm travel time. de Lima and Singh [26], using a non-linear kinematic wave model, compared the results of moving rainfall moving upwards and downwards on a synthetic plane surface with different velocities and showed that the sensitivity of runoff to the rain signal pattern decreases with an increase in storm velocity. Kim and Seo [27] studied the effects of storm movement in a synthetic V-Shaped catchment using a dynamic wave model and showed that the storm velocity and direction can be a governing factor in determining peak discharge and peak arrival time. Storm

movement is also shown to have further effects on other processes such as erosion [33,34] and sediment transport [35].

While some studies have previously focused on the effects of storm movement on runoff generation, most have focused on specific cases of real rainfall events, which mostly cannot be generalized for other rain events or catchments. It has also been previously pointed out that the effects of rainfall spatiotemporal variability on runoff are case-dependent and differ from catchment to catchment [36]. A critical reading of the literature highlights the need for further detailed studies and more evidence of the possible effects of storm movement and rainfall variability on runoff generation, and subsequently, a catchment's flash flood response. While observations and monitoring of real storm events may be key for a better understanding of these issues, modeling is also an essential tool. Observations are, by their own nature, constrained by the events which and when they occur. Therefore, replications and a comparative analysis are nearly impossible to conduct. In the case of moving storms, it is of course possible to monitor rainfall and runoff, but the same storm may not occur due to other velocities or directions, making direct comparisons impossible. Modeling is the only tool which systematically allows us to study the sensitivity of the system to variations in processes, and to make systematic observations of the response of the system to controlled forcing. Moreover, the analysis of the majority of previous studies has been limited to discharge and hydrographs, which may be insufficient as a flash flood severity metric. We extend most previous studies by not only focusing on resulting hydrographs but also considering additional flow parameters, such as water depth and flood extent, which provide a more useful comprehensive view of the impact of storm movement, taking the conclusions of the study a step closer to flood damage and risk estimation [37]. Although the availability of high resolution gridded data (radar or satellite) is increasing, still in most countries design rainfall hyetographs for a given duration and return period are used in designing stormwater drainage systems, flash flood and urban flood risk assessment, or proposing flood protection measures [38]. While the importance of rainfall movement has been highlighted in most previous studies, it is worthwhile studying to what extent the use of a rain hyetograph under a moving condition can affect rainfall/runoff and flash flood modeling results.

In this work, we focus on the effects of storm movement on surface runoff generation and how they manifest in flood response. We use the Kan catchment in Teheran, Iran as an example of an allochthonous system. We emphasize that the specific objective of the present study is not to represent nor reproduce a real flooding event, but instead to conduct a theoretical comparative analysis of the influence of moving vs stationary rainfall on flooding, via surface flow simulations. We also explore how the allochthonous nature of the river affects the response and how this influences the relationship between hydrographs observed at given points and flood response as a whole. To facilitate the assessment of the effects of rainfall movement, we make some simplifying assumptions. Infiltration and roughness, which strongly influence runoff generation and the resulting hydrographs, also introduce complexity and add uncertainty, thus reducing model interpretability [39]. Consequently, we construct a semi-idealized setup consisting of an impervious surface with low roughness to analyze model sensitivity to moving rainfall under different velocities and directions in comparison to stationary rainfall, although we also contrast with simulations considering infiltration and roughness. The methodology used in this study is mainly inspired by the downward approach [40,41] and has been previously used for studying the sensitivity of runoff generation to different processes in hydrodynamic modeling [42,43].

The Kan catchment has been chosen for this study for many reasons. Firstly, due to its steep topography, semi-arid climate, and rocky surface, the Kan catchment exhibits high potential for flash flood occurrences and has witnessed several flash flood incidents in recent decades [44–46]. Additionally, due to the catchments interesting and highly differentiated topography, rainfall in the catchment is mostly concentrated in the northern mountainous regions [46,47], which can result in flash-floods through the Kan river in the urbanized sectors of the capital city Tehran (Figure 1). Due to its topography and climate,

the Kan catchment offers an opportunity to study the effect of storm movement on the hydrodynamic modeling of flash floods.

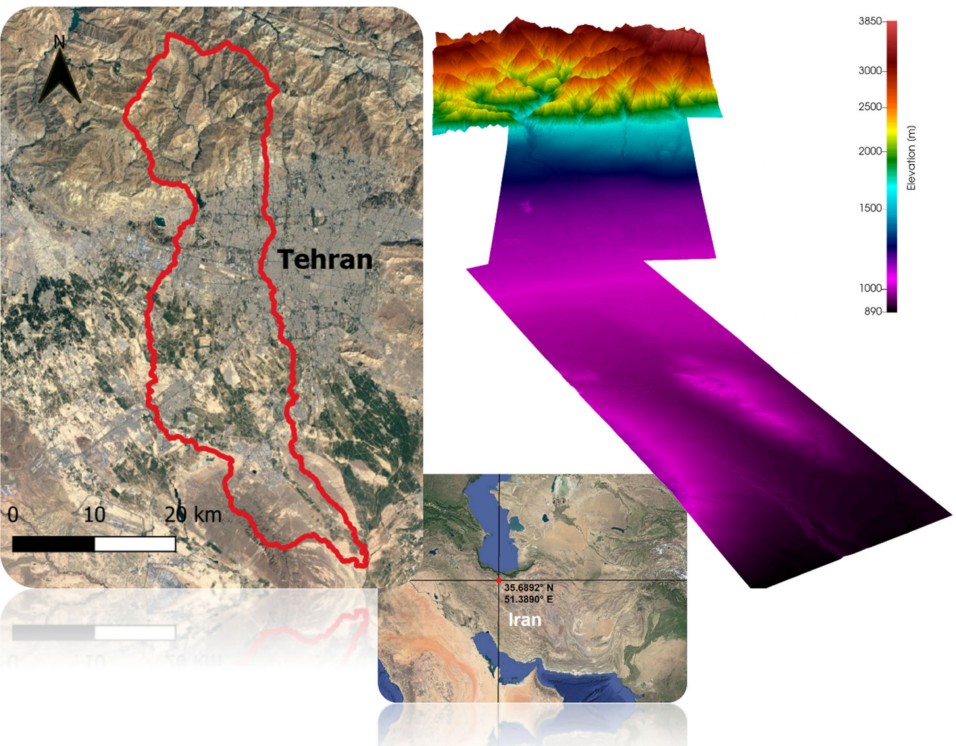

**Figure 1.** The Kan catchment. The red line shows the boundary of the catchment, noting that the flow is South of the Alborz Mountains range, just South of the Caspian Sea.

## 2. Materials and Methods

### 2.1. The Catchment

The study area is the Kan river catchment with an area of approximately 800 km², located in the district of Tehran (Iran). The catchment's elevations range from 900 to 3800 m.a.s.l. with an average slope of 30 degrees from north to south. The main stream in the catchment (Kan River), ranging from north to south, has an approximate length of 72 km. An overview of the boundary (left) and the topography (right) of the catchment is shown in Figure 1. The catchment area can mainly be divided into three main sections. The upper catchment area with a size of approximately 200 km² can be mainly specified by steep mountainous terrain with narrow river valleys. The surface of this part of the catchment is mostly covered by rock mass facies and rocky outcrop [44,47]. The middle catchment region with an area of approximately 230 km², mainly corresponds to the municipal area of Tehran city. The lower catchment section can be defined as a semi urban area consisting of patches of urban areas surrounded by agricultural fields. The surface of the middle and lower parts of the catchment are relatively flat in comparison to the upper region.

The Kan catchment exhibits a semi-arid climate [44,47,48] which is mainly characterized by low annual precipitation, large rainfall variability from year to year, clear skies, and low humidity [49]. This kind of climate is also known to exhibit complex patterns of rainfall spatial variability from year to year and even during single rainfall events [50,51]. The topographic characteristics of the catchment (i.e., elevation differences from south to north) are a governing factor in the annual rainfall and temperature in the area. Depending on altitude, the annual rainfall in Tehran city ranges from 145 mm in the southeast up to 422 mm in the north [52]. In another report, the annual rainfall for the upper section of the catchment has been reported to be 640 mm [53]. The area is reported to have had

an average wind speed of approx. 9.6 km/h between the years 1950–2006 [54] and the dominant wind direction in the area is reported to be west to east [55,56].

## 2.2. Hydraulic Model

The core hydrodynamic process in the mathematical and numerical model is free surface flow. Flow generation is modeled using the hydrodynamic module of ProMaIDes-Protection Measures against Inundation Decision support (https://promaides.h2.de/promaides/, accessed on 7 June 2022). ProMaIDes is a software package for risk-based assessment of flood protection measures for river floods and storm surges [57]. The core hydrodynamic solver of ProMaIDes [58] solves the Zero-Inertia (ZI) approximation (Equation (1))-also often named as the diffusive wave approximation-to the shallow water equations [59]:

$$\frac{\partial h}{\partial t} + \nabla \left( \frac{h^{5/3}}{n\sqrt{\|Z\|}} Z \right) = R \tag{1}$$

in which $h$ is water depth [m], $t$ is time [s], $n$ represents Manning's roughness coefficient [$ms^{-1/3}$], $Z$ refers to the water surface gradient [-], and $R$ [m/s] is a source or sink term, which in our case in the 2D domain represents rainfall intensity and, in the 1D domain, $R$ corresponds to the lateral inflow for each computation reach. The ZI equation has been widely used and evaluated in runoff modeling [43,60–62] and has been shown to be valid for rainfall-runoff simulation [60,63].

ProMaIDes is able to apply the solution of the ZI equation in two types of domains: 2D overland flow and 1D stream channel flow. A first-order finite volume scheme with implicit time integration [58] is used for both 1D and 2D flow domains. The 2D domains are discretized with regular grids and the 1D streamflow domains are discretized by cross-sections along the stream. The 1D and 2D domains are coupled together explicitly in time, by means of mass flux exchange at specified coupling time intervals. The 1D–2D mass flux exchange is formulated by means of a lateral weir equation in which the exchanged volume is governed by surface level differences [58]. The 1D–2D hybrid approaches are known to perform well for hydrodynamic flood modeling [62,64–66]. Additionally, ProMaIDes also allows for a coupling between 2D–2D subdomains for further computational efficiency, a feature which has been used in this study (Figure 2).

The simulations were run with ProMaIDes on an Intel Core i5-10400F CPU@2.90 GHz.

## 2.3. Study Setup

### 2.3.1. Model Setup

This study was conducted using a spatially distributed 1D–2D hydrodynamic model. Three subdomains (represented in 2D) were coupled with 14 river models (represented in 1D), which are shown in Figure 2. Topography in the 2D domains is based on the TanDEM-X 12 m digital elevation model. A cell resolution of dx = dy = 50 m is used for the upper subdomain (which represents the mountainous section of the catchment) and for the two lower subdomains a higher resolution of dx = dy = 25 m is used. The elevation of each cell was obtained by arithmetic averaging of the original 12 m resolution DEM. Altogether, the three 2D computational domains consist of a structured grid of approximately 1.4 million square cells. Regions in the subdomains which are located outside the catchment boundary (Figure 1, red line) are marked as non-computational cells.

Fourteen river models with a total of 1075 cross sections (with average separation of 150 m) have been set up for the 1D model. Cross sections elevation is based on the TanDEM-X 12 m DEM. Cross section elevation is resampled into points at every 5 m by bi-linear interpolation. With the help of satellite images and DEM, an intensive plausibility check of the cross sections was done.

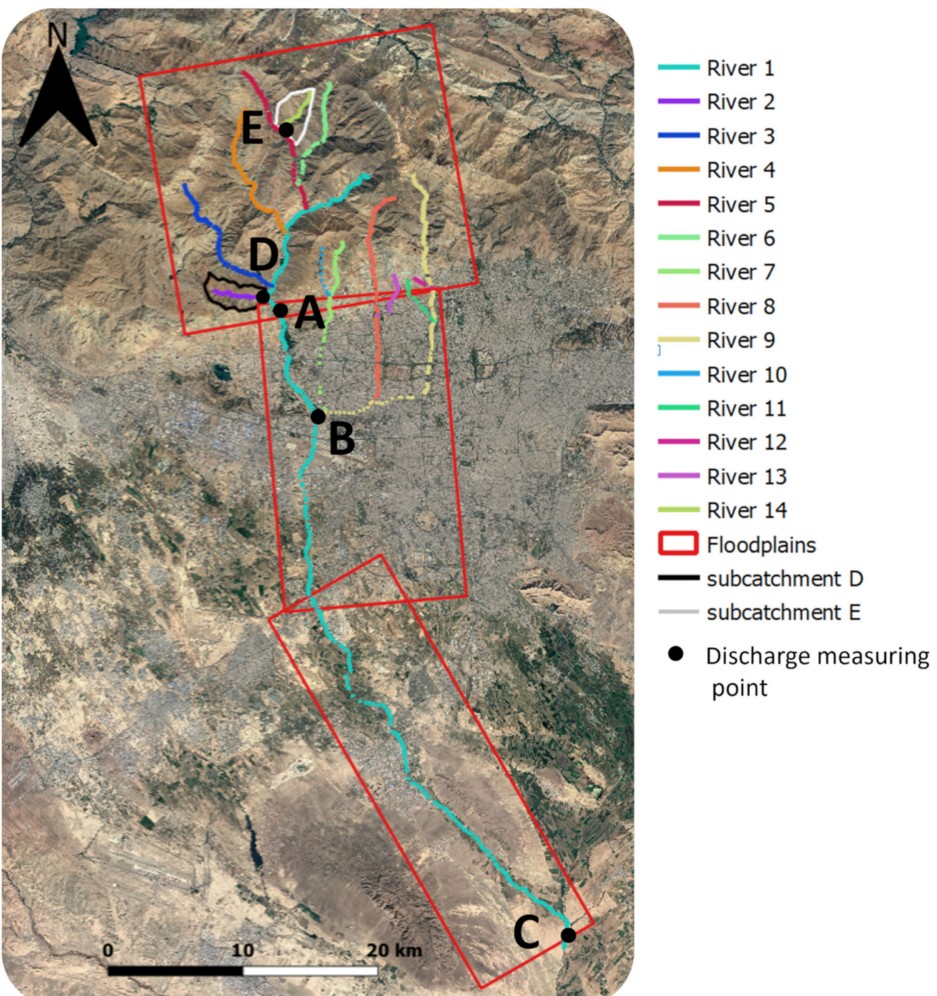

**Figure 2.** The 1D–2D hydrodynamic model for the Kan catchment. Discharge is measured at 5 locations during the simulations. Point A (end of the upper subdomain) and C (end of the lower subdomain) are located on the Kan River itself, whereas point B is just upstream of the confluence of a tributary of the Kan River (i.e., does not gauge the discharge of the Kan River itself). Points D and E are located at the end of two sub-catchments in the upper domain.

We calculated discharges at five different locations during the simulations. Measuring points A and C (Figure 2) are located in the main Kan river. Measuring point B is at the end of River 9 (Figure 2), which is one of the main tributes to the Kan river. It is interesting to point out that B is located at a position in which the contributing streams have over half of their channels in the urban floodplain. Additionally, in order to see the effects of rainfall movement also on a sub catchment level, two sub catchments in the upper reaches of the area were chosen (illustrated in Figure 2). We named the sub catchment in the southwest part of the upper catchment (black line) sub catchment D and the other located in the northern part (white line) sub catchment E. The main criteria for choosing the sub catchments is the direction of the main channel in the sub catchments: west-to-east in D, and north-to-south in E. Simulated discharge is measured at the end of the main stream of each sub catchment, also shown in Figure 2.

As mentioned before, inspired by the *downward approach* [40,41], we first construct an idealized model for this study. In previous flood modeling literature, infiltration has been either neglected [67–69], substituted by an initial abstraction in the rainfall [70,71], computed using dedicated equations such as Green–Ampt [72,73], Horton [74,75], or a simplification of the former equations [76]. The surface in the upper reaches of the catchment is mainly characterized by homogeneous rocky surfaces and an inceptisol

soil type [47,77] which is known to yield very limited infiltration rates [78]. The central and lower sections of the catchment contain the Metropolitan area of Tehran, which is characterized by highly dense urbanized regions with surfaces yielding very low infiltration rates [11], hence favoring runoff.

In this study, to avoid extra arbitrary assumptions and to keep the focus on the primary aim of this theoretical exploratory study, we take the simplifying assumption of neglecting infiltration rates, resulting in an idealized setup with the assumption of an impervious surface. This setup also allowed for the results of the simulations to only demonstrate differences caused by rainfall movement in the simulations. Additionally, a homogeneous roughness with the Manning's coefficient of 0.01 s/m$^{1/3}$ is set, considerably smoother than the actual catchment surface. This is close to a frictionless setup and not supported by the Zero-Inertia equation [59]. This type of idealized setup has been previously used for studying the sensitivity of runoff generation to different processes in hydrodynamic modeling and is argued to facilitate the understanding of the roles of different processes and catchment features in runoff generation [42,43]. Further processes such as evaporation and subsurface flow are neglected while they occur at a different temporal scale. This idealized setup enables the simulation results to highlight the effects of rainfall movement in the resulting hydrographs and ease interpretation. Additionally, we also contrast the results of these simulations with simulations considering infiltration and heterogeneous roughness.

2.3.2. Rain Modeling

Rain is synthesized only to the upper subdomain of the model in order to create an allochthonous river flooding scenario in the central and lower parts of the catchment which contain the urban areas. This is also in line with the general climate of the area with reported high annual precipitation in the north mountainous reaches of the catchment while having very low annual rain volumes in the central and lower sections [46,47,52].

Fifty synthetic rain signals were generated for this study. We distinguish the rain signals based on their intra-storm temporal variability, described by the variance ($\sigma^2$). Intra-storm rainfall distributions are generated using a microcanonical random cascade model [79] with hourly resolution. The main principle of the model is to increase the temporal resolution of a certain volume gradually while preserving general statistics and mass in every stage. The required statistical parameters for the disaggregation are calculated by stepwise aggregating observed hourly timeseries to coarser temporal resolutions. Detailed information on model structure and the parameterization procedure can be found by Pohle et al. [79].

Although a few rain gauge stations are present in the catchment [80], the recordings of these gauges are not publicly accessible and could not be used for this study. The only publicly accessible rainfall data of the catchment are IDF curves presented by Yazdi et al. [80] which are derived from the rain gauge recordings in the catchment. Due to this lack of local precipitation data and difficulties in parameterizing the rainfall model for the Kan catchment, a location in CA, USA (Stockton, CA, USA) with a similar type of cold semi-arid climate (BSk) as Tehran city was chosen. Hourly recorded rainfall recordings from the mentioned area (Stockton, CA, USA) for a duration between 1975 to 2020 were used as input for the rainfall model parameterization. Based on IDF curves offered by Yazdi et al. [80], an 8 h 10 mm/h continuous rain event (Figure 3a) is known to be a 100-year return period event in the Kan catchment. Consequently, using the random cascade model, we disaggregated an 80 mm precipitation depth into 8 hourly timesteps to generate 49 plausible rain signals. These signals, in addition to the simplified continuous signal (Figure 3a), comprise the 50 rains which were used for this study. Figure 3 shows nine of these hyetographs. It is worthy to re-emphasize that the specific purpose of this study is not reproducing a real-world flooding event for the Kan, but instead to conduct a sensitivity analysis on the influence of rainfall movement on surface flow and flooding in hydrodynamic flood modeling. Therefore, the selection of the rainfall input is somewhat arbitrary, given that it remains plausible and representative of real situations.

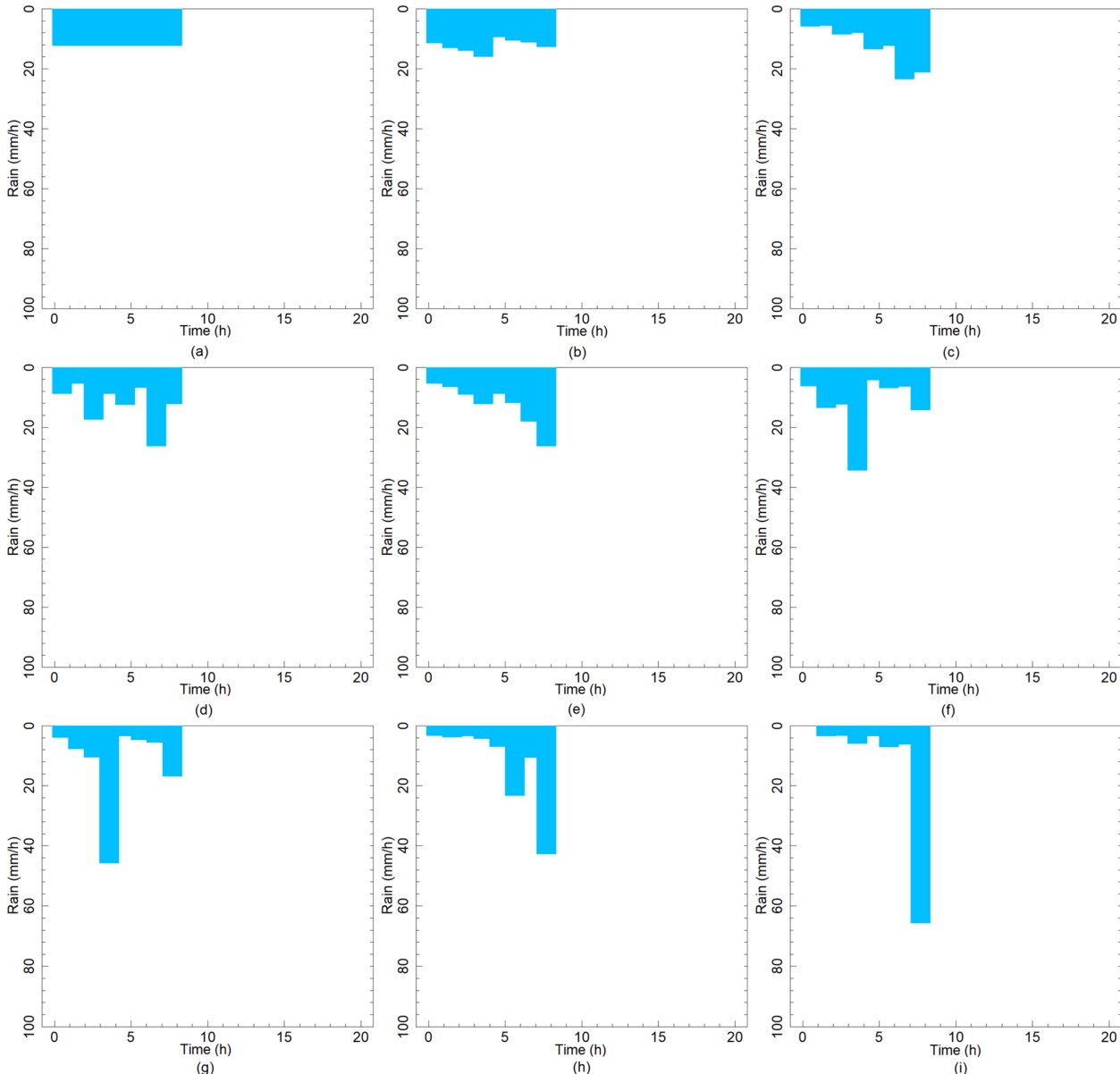

**Figure 3.** Nine of the generated synthetic rain signals illustrated as examples. Rain signals $\sigma^2 = 0$ (**a**), 4.55 (**b**), 48.67 (**c**), 49.04 (**d**), 50.34 (**e**), 99.22 (**f**), 214.50 (**g**), 208.76 (**h**), 496.41 (**i**) are presented.

We emphasize that the total rainfall depth in all generated rain signals is equal, making them fully comparable to one another. The generated rains have variances ($\sigma^2$) ranging between 0 to 681.6. In general, rain signals with a higher variability (e.g., Figure 3h) tend to have higher peaks in comparison to lower variability rains (e.g., Figure 3a). An overview of the rain peak intensities, peak timings and also a comparison on the volume of the first and second half of each rain signal are shown in Figure 4. It is seen that the peak intensities of the rains show a monotonic relation to $\sigma^2$ while this relation is approximately sigmoidal in the case of time of peak intensity and exponential in the case of the volume distribution.

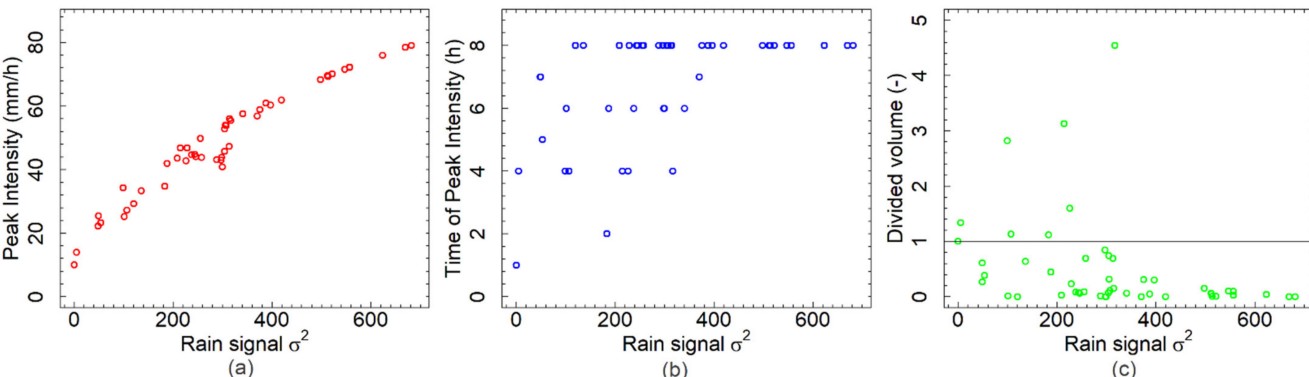

**Figure 4.** Peak intensities (**a**), time of peak intensity (**b**) and the first half volume of each rain divided by the second half volume of the rain (**c**), for all 50 rain signals used in this study.

To enable rainfall movement in the model, the rain area (upper subdomain) is discretized into rectangular elements with a width resolution of $dw$ = 0.5 km (Figure 5). The rain pulse starts in each rectangle at a specific time in the simulation, given by:

$$t_n = (n-1) \times \frac{dw}{v} \tag{2}$$

where $t_n$ is the time when the rainfall starts in rectangle $n$, $n$ is the rectangle number (e.g., 1, 2, 3 …) starting from the direction which the storm enters the domain, $dw$ represents the domain discretization resolution (in our case 0.5 km) and $v$ is storm velocity. An example of this procedure is shown in Figure 5. As seen in Figure 5a, for a storm moving from west to east with a velocity of 5 km/h, the rain pulse starts in the west-most rectangle at $t$ = 0 min. The 5 km/h storm velocity leads to the rain reaching the second rectangle at $t$ = 6 min (Figure 5b). Subsequently, the rain starts in the third rectangle from the west at $t$ = 12 min (Figure 5c) and so on. Further on the hyetographs of a few rectangles from the west are explementary illustrated in Figure 5d,e. Storms moving from east to west are analogous. Analogously, for storms moving along the north-south axis vertical directions, the same discretization is dl = 0.5 km.

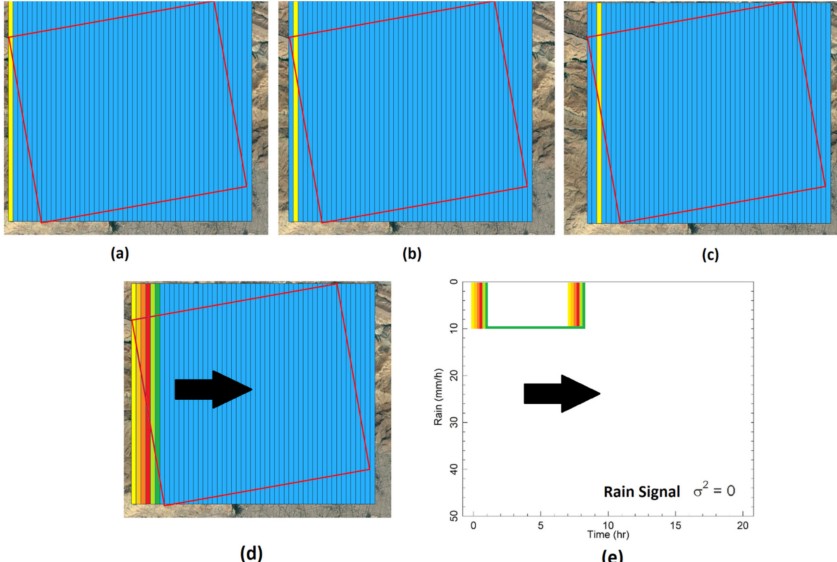

**Figure 5.** Discretization of the upper subdomain (red line) into cells (blue rectangles) with $dw$ = 0.5 km to model rainfall movement in the simulations. As an example, the movement of rain signal $\sigma^2$ = 0 with a velocity of $v$ = 5 km/h from west to east for some cells is illustrated (**a–c**). The hyetographs of the few first cells (illustrated in **e**) from the west (**d**). The colors in (**d**) and (**e**) indicate the cell.

During all simulations, all cells in the simulation receive the same rainfall volume, but also the same rainfall hyetograph, only displaced in time. To further explain how rainfall movement is represented in the simulations as an example for the constant pulse ($\sigma^2 = 0$), regardless of the speed of the movement, each of the discretized rectangle receives a rain intensity of 10 mm/h for 8 h. Only the times when the rain starts in each of the rectangles differ, depending on the velocity of the storm. This is in line with Yen and Chow [25] which defined rainfall movement as storms moving at different speeds over the area despite having the same rainfall duration and volume at every point of the watershed. Although this definition may be conceptual and may be a simplification of actual storm movement in reality, it allows for a systematic study of the effect of storm movement on runoff and flooding through a modeling approach and also enables a plausible comparison among different storm velocities and directions. We emphasize that all the cells in the simulations under all velocities and directions and also in the stationary rain scenarios receive the same volume of rainfall during the whole simulation. Consequently, the rainfall scenarios are fully comparable in terms of rainfall volume. It has been previously pointed out that contradictory conclusions in assessing the effects of rainfall variability between different studies may be related to the difficulties in guaranteeing consistent rainfall volumes [81].

Another definition of rainfall movement was also proposed by Ogden et al. [82], who studied moving storms of different speeds and different rainfall intensities proportional to the storm movement speeds. However, after testing this rainfall movement definition, we decided to avoid using it in this study due to different rainfall velocity-dependent intensities dominating the main shape and peak of the hydrographs and causing the results to be incomparable with one another. Some examples of simulated hydrographs based on this definition are shown in Figure S1 in the Supplementary Materials. The definition provided by Yen and Chow [25] was chosen for this study, which enabled a plausible comparison among different movement setups and allowed the results to illustrate the differences caused by different movement velocities of the same rain hyetograph.

Velocities of $v = 5$, 10, and 25 km/h were chosen for this study, with addition of a stationary rainfall over the catchment. The chosen velocities are in line and close to the average wind speed of 9.6 km/h, reported from the area [54]. Additionally, for each velocity, four movement directions were considered. Directions from east to west, west to east, north to south and south to north. Fifty synthetic rain signals, in combination with 3 velocity setups in addition to stationary rainfall setups resulted in a total of 200 scenarios. In addition, simulations were also run with some selected rains under 4 different directions. The simulations were run for a 24 h time period.

## 3. Results and Discussion

The runoff hydrographs at measuring point A (the outlet of the mountainous region, see Figure 2) for 4 of the generated rains (Figure 3a,b,e,f) under the different velocity setups with a direction of west to east (dominant wind direction reported for the area) are shown as an example in Figure 6. The four selected rains have noticeably different hyetograph variances. Differences are also present in the peak intensity and its timing for the selected rains with two signals having a peak intensity in the early stage and one in the end of the storm. The overall shapes of the hydrographs are mainly governed by the rainfall signal. Rains with higher temporal variability achieve higher peak discharges (Figure 6c,d), which is the result of the higher peaks in the corresponding rainfall hyetographs (Figure 4a).

Moving and stationary storms differ both in peak discharge values and the rising limbs of the hydrographs. The stationary rainfall is considered as the reference case. The different storm velocities result in peak discharges of different magnitudes and with different timings in contrast to the reference case. These differences increase with rainfall temporal variability ($\sigma^2 = 50.34$ and $\sigma^2 = 99.2$) as seen in Figure 6c,d. For some cases such as the $v = 5$ km/h rainfall, the peak clearly occurs at different times in comparison to the other velocities. As an example, the $\sigma^2 = 99.2$ rain with 5 km/h velocity shows a peak discharge at approximately $t = 350$ min, while the stationary rainfall setup reaches a peak

at around $t$ = 220 min. In general, moving rainfall tends to produce lower peak discharge values in comparison to stationary rain. Peak discharge reduces for slower storms. The rising limbs of the hydrographs are clearly affected by the movement. Stationary rain tends to have the fastest rise in comparison to moving rains. Slower storms also result in delayed onset of runoff.

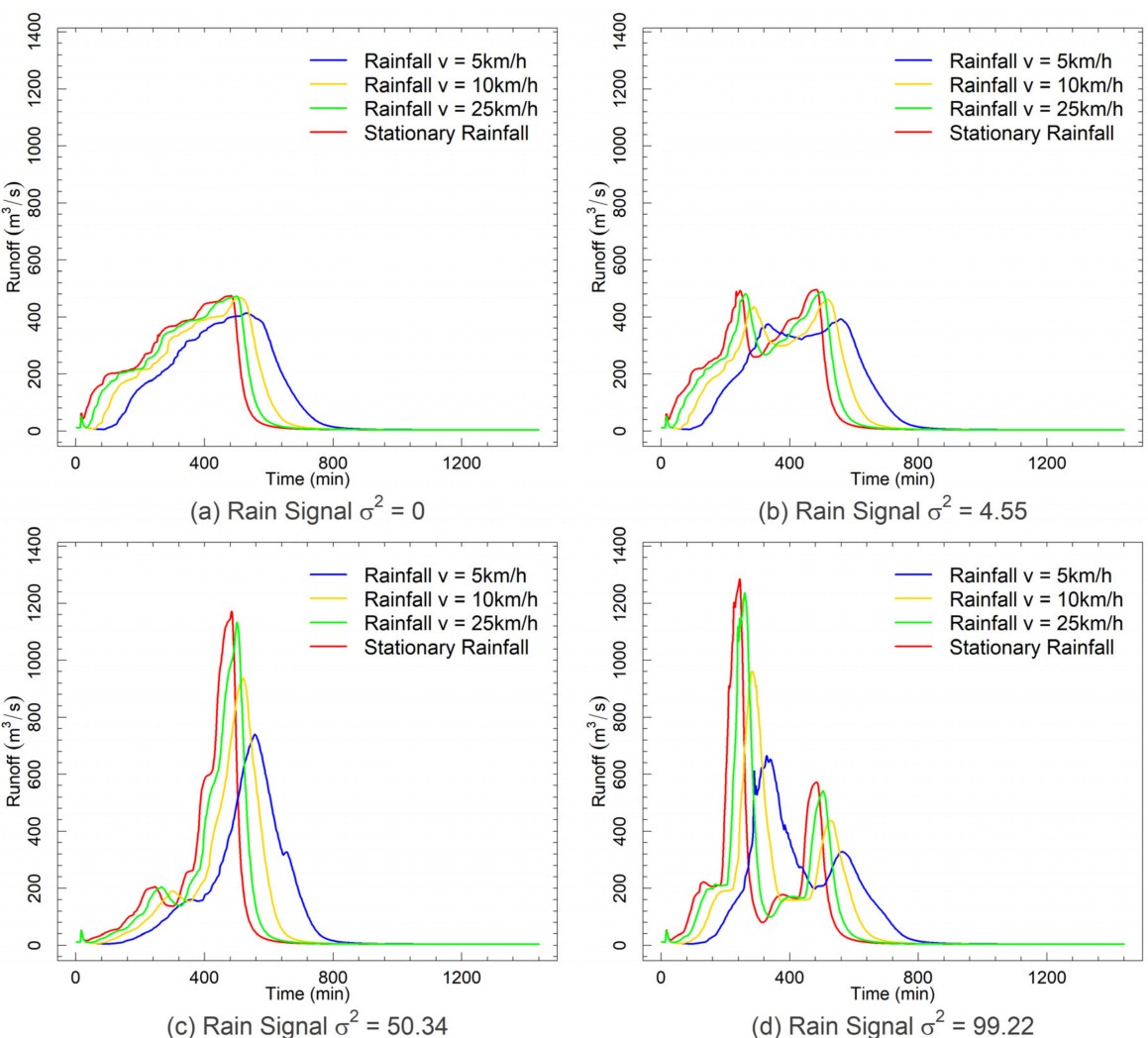

**Figure 6.** Hydrographs (at measuring point A) for four generated rains (Figure 3a,b,e,f) under stationary and moving setups with velocities $v$ = 5 km/h, $v$ = 10 km/h, $v$ = 25 km/h with a direction from west to east.

The hydrographs for the aforementioned rains moving from west to east under different velocities at measuring points B, C, D, and E (Figure 2) are shown in Figure 7. For these points, the aforementioned differences among different storm movement setups seen in measuring point A, hold. However, peak discharge magnitudes are less affected and are almost equal at the sub catchment measuring points (D and E) and only the rising limbs of the hydrographs are delayed by slower moving rainfalls. This is likely a scale-dependent response: small(er) sub catchments require smaller travel times to reach their peak discharge than the larger (full) catchment. It is also seen that, in measuring points B and C, highly variable rain signals (e.g., Figure 7h) tend to produce more differences in peak discharge estimated in comparison to low variability rains (e.g., Figure 7e).

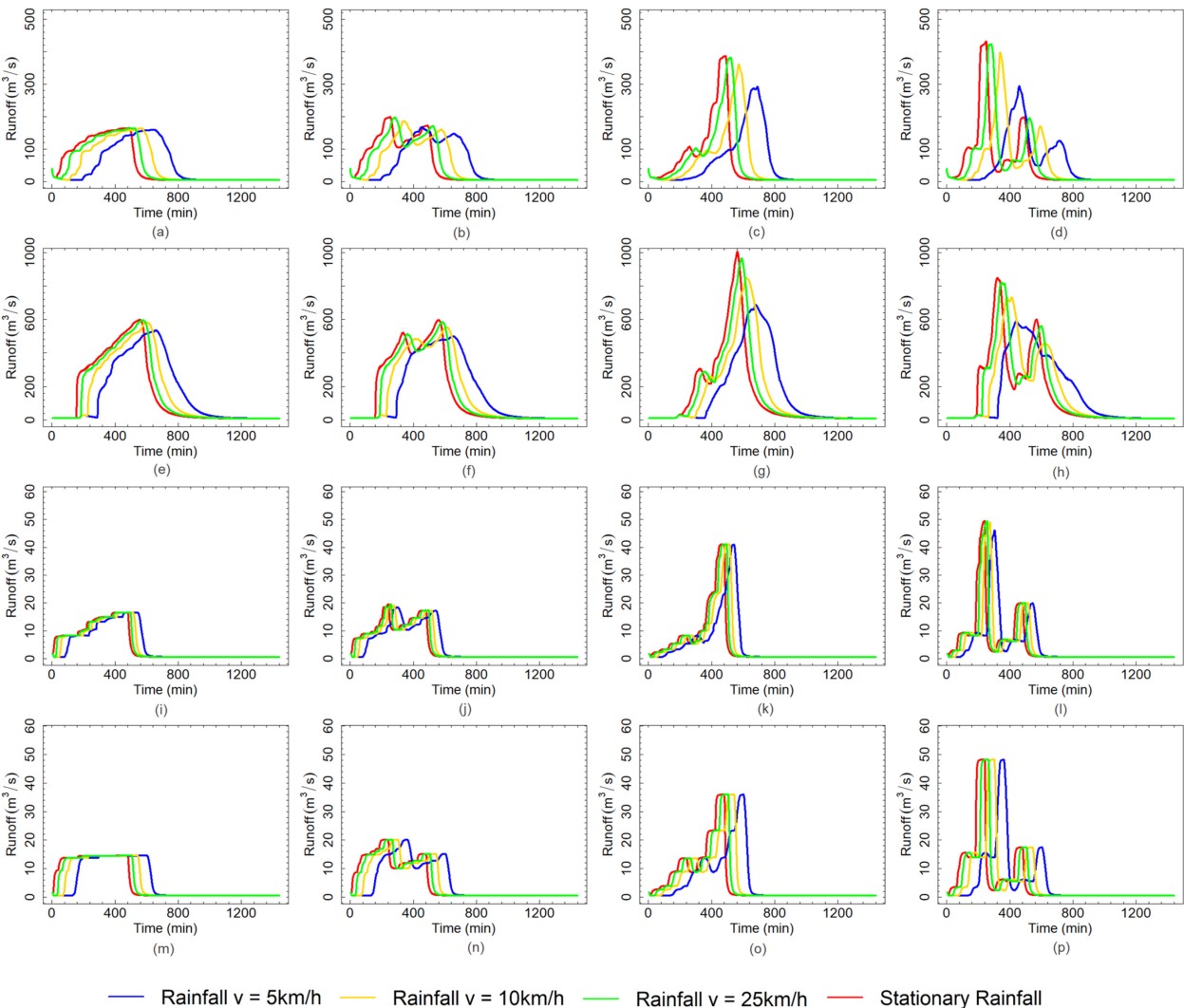

**Figure 7.** Hydrographs for four generated rains (Figure 3a,b,e,f from left to right respectively) under the stationary and moving setups with velocities $v$ = 5 km/h, $v$ = 10 km/h, $v$ = 25 km/h with a direction from west to east at measuring points B (**a–d**), C (**e–h**), D (**i–l**), E (**m–p**) (Figure 1).

We now turn to analyze the full ensemble. Figure 8 shows peak discharge for each rain signal under different moving setups normalized by the peak discharge of the stationary scenario of that rain, for each measurement point. The absolute peak discharge values for the stationary rainfall setups at all measuring points are shown in Figure 8f. In Figure 8f, it can be seen that the measured peak discharges at measuring point A show a clear dependency on hyetograph variance. Higher variances generally result in higher peaks. Peak discharges in points B and C show an asymptotic behavior with the peaks increasing up to a hyetograph variance of approximately 200 and then remaining relatively unchanged by hyetograph variance. At points D and E, the peaks are relatively constant and unaffected by hyetograph variance, which can be explained by the sub catchments' scale effect on the generated runoff.

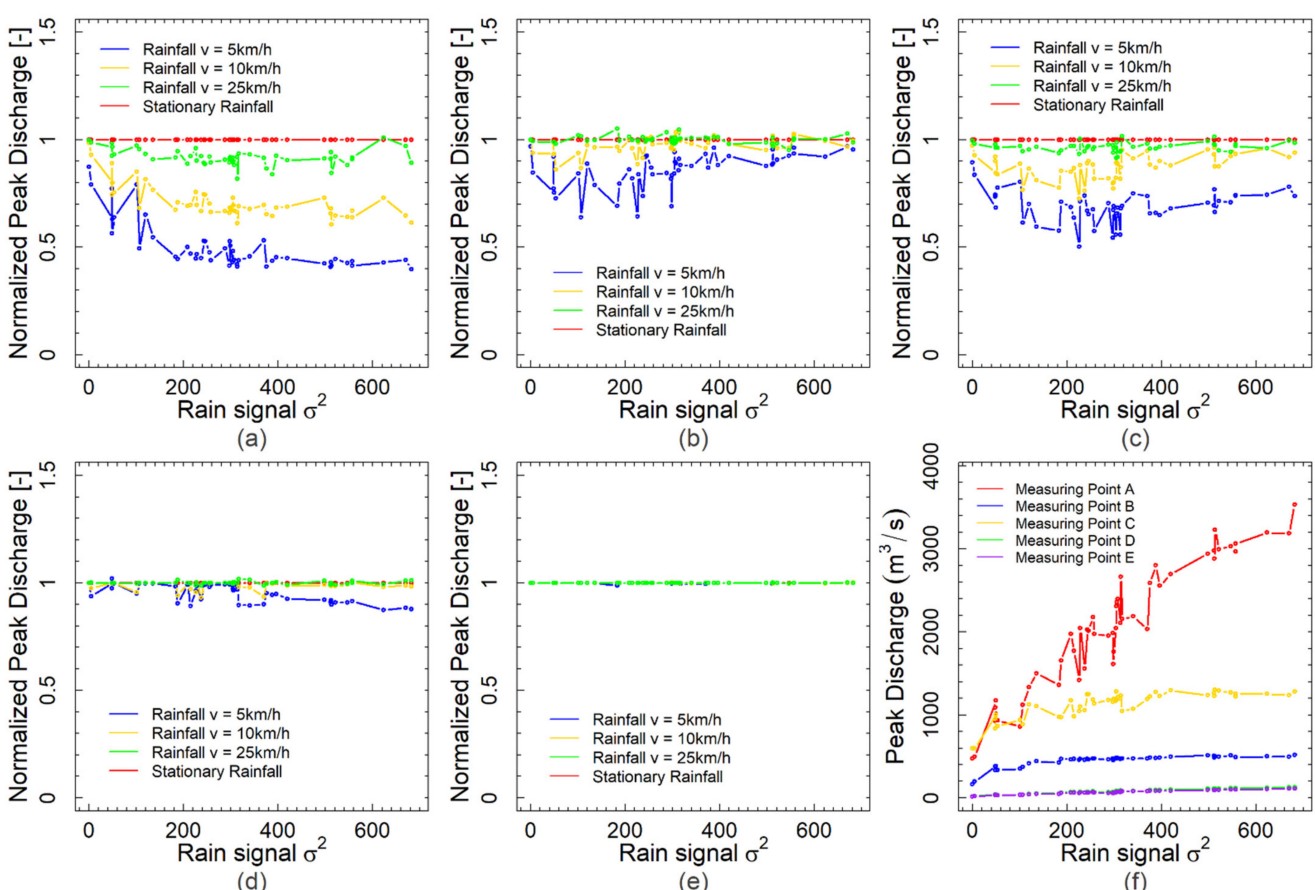

**Figure 8.** Peak discharge normalized by the peak discharge of the stationary scenario for rains under the stationary and moving setups with velocities $v$ = 5 km/h, $v$ = 10 km/h, $v$ = 25 km/h with a direction from west to east at measuring points A (**a**), B (**b**), C (**c**), D (**d**) and E (**e**). The absolute peak discharge values of the stationary rain setups for the five measuring points are shown in (**f**).

It is seen that for almost all cases, slower storms result in lower peak discharge estimates compared to faster moving storms. At measuring point A (Figure 8a), peak discharge differences arising from storm movement increase with rainfall variability, up to a variance of $\sigma^2 = 200$, and remain roughly constant for higher variance. This behavior can also roughly be seen in measuring point C (Figure 8c). In contrast, at measuring points B and C, the peak discharge differences arising from storm movement decrease with increasing variance in the high variability range ($\sigma^2 > 200$), with differences of less than 10% for the highest variated rains (e.g., $\sigma^2 \sim 680$) at measuring point B (Figure 8b). This can be compared to the same rain variability at measuring point A with differences reaching up to 60% (Figure 8a). This clearly indicates the important role of the catchment streams and floodplains located in the non-rain induced areas, which act as a buffer, greatly filtering rainfall variability and, to some extent, the effects of storm movement on discharge generation, lowering the relevance of storm movement on peak discharge estimates. This highlights how the allochthonous nature of a river may interact with rainfall variability and further on the effects of storm movement on runoff generation. Further on it can be seen that stationary rainfall in almost all cases results in the highest peak discharge. This is followed by faster storms which indicates that an increase in storm velocity leads to an increase of peak discharge in the streams. Only in a few cases and only at measuring point B (Figure 8b) does the fastest moving storm ($v$ = 25 km/h) generate a higher peak discharge than the stationary rainfall setup.

Clear differences can also be seen in peak discharge estimates among catchment and sub catchment levels. While the catchment measuring points (A–C) show big differences in

calculated peak discharge among different storm velocities (up to 60%), at a sub catchment level, the measured peak discharge values are almost identical or with much smaller differences for most cases (Figure 8d,e). As mentioned before, this is a scale-dependent response, in which small(er) sub catchments require smaller time travel times to reach their peak discharge than the larger (full) catchment.

The time to peak discharge for each rain signal under different moving setups, normalized by the time of peak discharge of the stationary scenario of rain at all five measuring points, is shown in Figure 9. In general, slower storms generate higher delays in peak discharge in comparison to faster storms. Stationary rains result in the fastest peak discharges. This can be explained by the whole domain receiving rainfall from $t = 0$ in the stationary rainfall simulations, while in the moving rainfall cases, a time is required for the storm to cover all the domain, which results in a delay in peak discharge. Due to this, faster storms, which cover the domain in a shorter time compared to slower storms, result in a faster peak discharge in comparison to slower moving storms. The differences in the delays are location dependent. At measuring points A, C, and D, the differences are less in comparison to measuring points B and E. The large differences in points B and E can be explained due to their location in the catchment. Point E measures the flow resulting from sub catchment E, located in the north central part of the catchment. Due to the storm moving from west to east, the time required for the storms to reach the sub catchment is seen as a delay in peak discharge (relative to the same time datum), while in the stationary setup the rain instantaneously starts in the sub catchment, resulting in a much earlier peak discharge. The same reasoning can be used for point B, which is located at a tributary to the main Kan river, and its flow is mainly sourced from the eastern part of the catchment. On the other hand, smaller differences are seen (Figure 9d) for measuring point D in sub catchment D (Figure 2), which is located in the western part of the upper catchment sub domain.

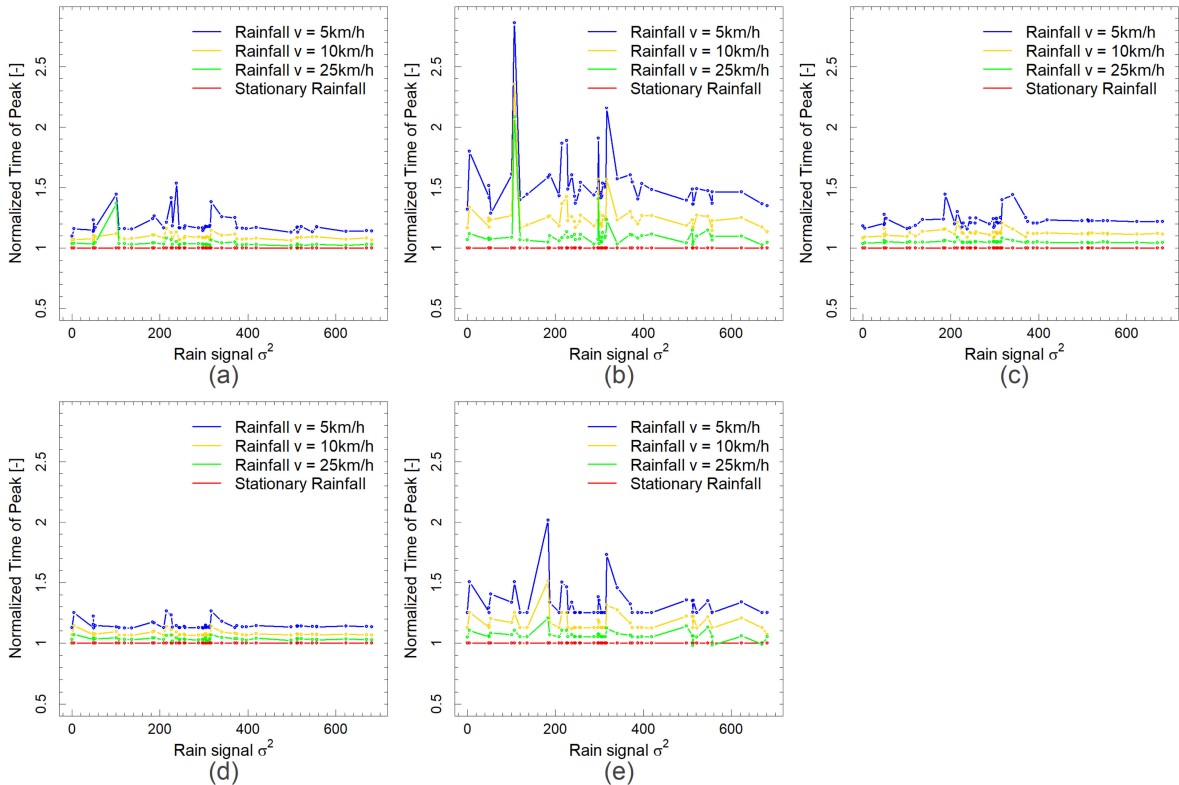

**Figure 9.** Normalized time of peak discharges by the time of peak discharge of the stationary scenario for rains under the stationary and moving setups with velocities $v = 5$ km/h, $v = 10$ km/h, $v = 25$ km/h with a direction from west to east at all measuring points A (**a**), B (**b**), C (**c**), D (**d**), E (**e**).

Figure 10 shows the spatial distribution of the envelope at maximum water depths (maximum water depth at any time for a given location) for $\sigma^2 = 99.2$ (Figure 3e) rainfall. The figure compares stationary rain (Figure 10 left) and moving rain with a velocity of 5 km/h with a direction from west to east (Figure 10 right). During both simulations, water overflows the river channel, causing flooding at several points. Most of the flooding occurs in the lower part of the catchment in the surrounding areas of the main Kan river. Some flooding can also be seen in the middle floodplain, mainly from rivers 7 and 8 (Figure 2). The $v = 5$ km/h rainfall setup tends to produce smaller flood areas compared to the stationary rainfall setup. This is mainly due to the higher peak flow values of the stationary scenario (Figure 8) which consequently result in higher water levels and higher outflow into the floodplains.

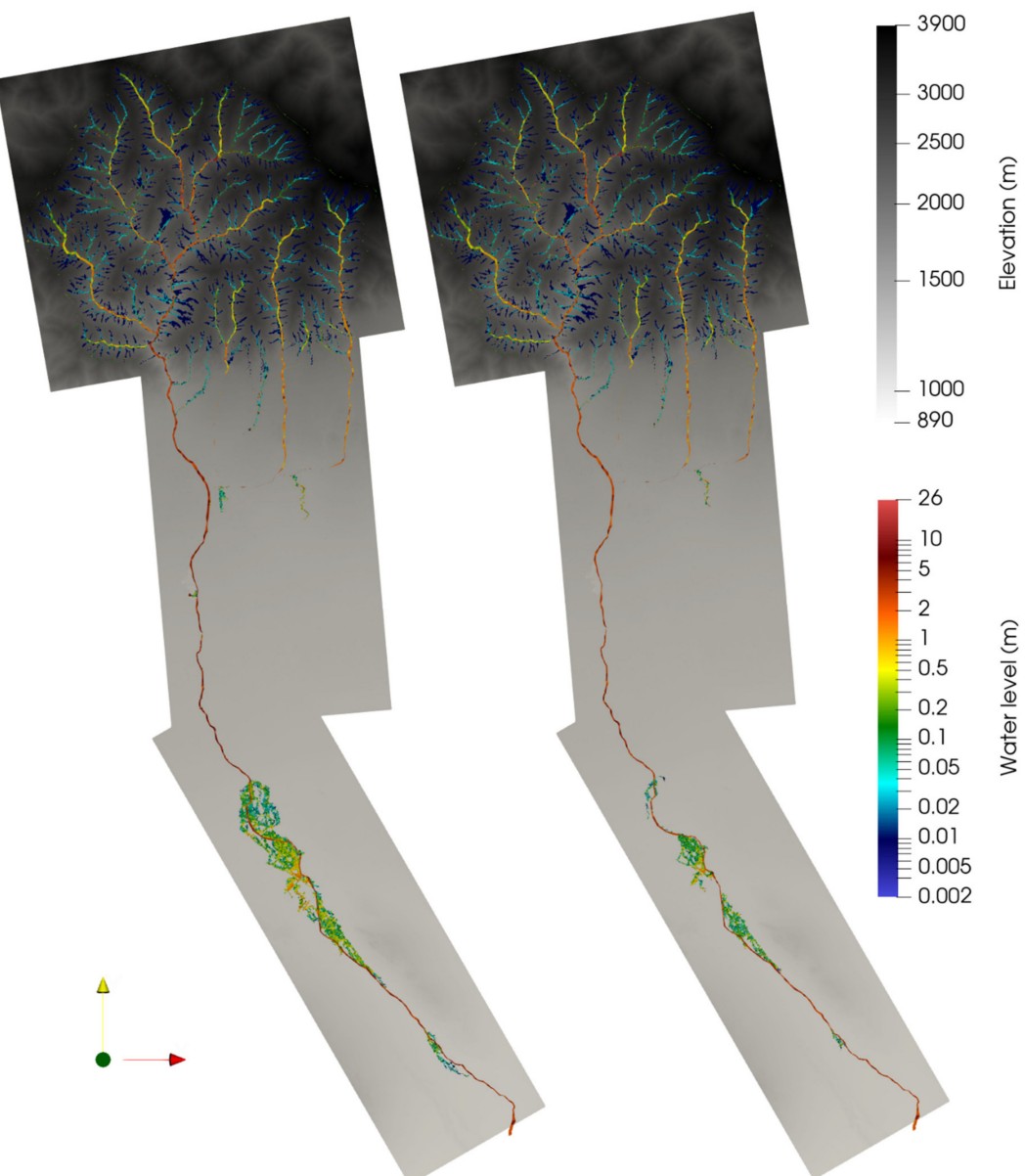

**Figure 10.** Spatial distribution of the maximum water depths for the $\sigma^2 = 99.2$ rainfall (Figure 3e) under the stationary rainfall (**left**) and a moving rainfall with $v = 5$ km/h with a direction from west to east (**right**).

Since discharge itself is not necessarily a metric indicating the severity of a flash flood, especially in an allochthonous setting, absolute flooded areas (Figure 11a–d), and also

flooded areas normalized by the flooded areas of the stationary rainfall setup (Figure 11e–i), are shown in Figure 11. The figure presents the flooded areas of the two lower subdomains. Recall that these regions do not receive rainfall, and therefore the flash-flooding can be considered allochthonous. The subfigures indicate the flooded area exceeding a particular water depth (exceedance categories), which allows for a further comprehensive analysis than the total flooded area. From the absolute flooded areas (Figure 11a–d) it is seen that, in general, with an increase in rainfall variance, the flood magnitude increases. In the majority of the cases, stationary rainfall results in higher flooded areas in comparison to moving storms. Only in some cases with high velocity storms ($v$ = 25 km/h), moving storms produce higher flooded areas in comparison to stationary. Faster storms generally result in higher flooded areas in comparison to slower storms. The normalized Figure 11e–i give a comparative view of the results. In general, hyetograph variance does not clearly influence the effect of storm movement, with the differences in flooded areas between different storm velocities being relatively similar. The behavior of the normalized areas for different flood depth thresholds are rather similar, showing that the differences in the absolute values is simply a matter of scaling. Slower storms with $v$ = 5 km/h and $v$ = 10 km/h result in lower flooded areas up to approximately 60% and 40% in comparison to stationary rainfall, respectively. Both the absolute flooded areas and also the normalized figures confirm that storm movement can have significant effects on flooded areas.

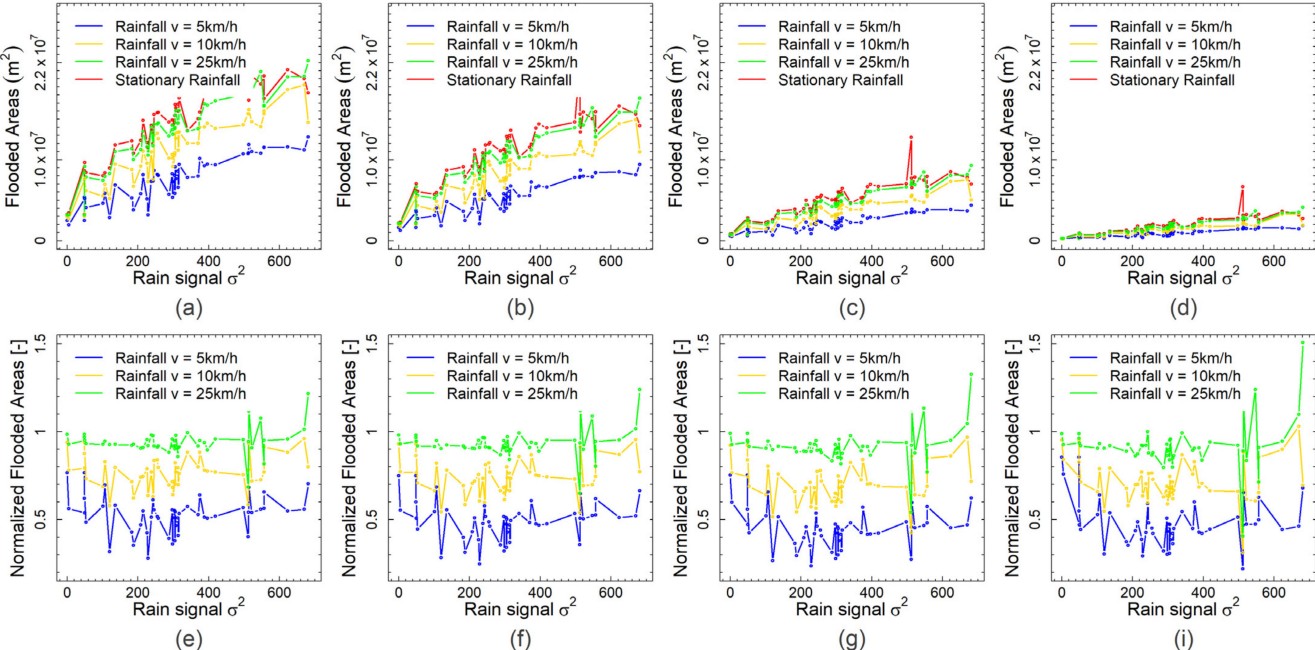

**Figure 11.** Flooded areas (**a–d**) in the two lower floodplains for all rains in the stationary setup and moving setups with velocities $v$ = 5 km/h, $v$ = 10 km/h, $v$ = 25 km/h with a movement direction from west to east. (**e–i**) present the flooded areas of the moving setups normalized by the flooded area of the stationary scenario. Figures from left to right represent water levels above 10 cm, 20 cm, 50 cm and 100 cm respectively.

We turn now to storm direction. Hydrographs at measuring point A resulting from different incoming storm directions are shown in Figure 12, for two rainfall variabilities ($\sigma^2$ = 0 & 99.2). Each figure represents one velocity and line colors represent the movement direction. Stationary rain has also been plotted in the figures as a reference case. In general, storm direction produces differences in hydrograph shapes, specifically in slower moving rainstorms. As the velocity of the storm increases, the differences in the hydrographs are reduced. The storm moving from north to south (blue line), which is in the main stream direction of the main Kan river, tends to produce the highest peak discharge compared

to other directions. Storms moving from south to north (green line) result in the lowest peak discharge. This is mainly in line with de Lima and Singh [26], which compared moving rainfall moving upstream and downstream on a synthetic plane using a non-linear kinematic wave model and showed that downstream moving rain produces higher peak discharge in comparison to upstream moving rain. It can also be seen that at lower velocities, higher rainfall variability (Figure 12d) results in larger differences in peak discharge for all directions with regard to stationary rainfall, in contrast to lower rainfall variability (Figure 12a). At higher velocities, the differences among the hydrographs tend to settle out, reducing the significance of storm direction on the resulting hydrographs. This behavior is similar at the other measuring points in the catchment between different directions of movement.

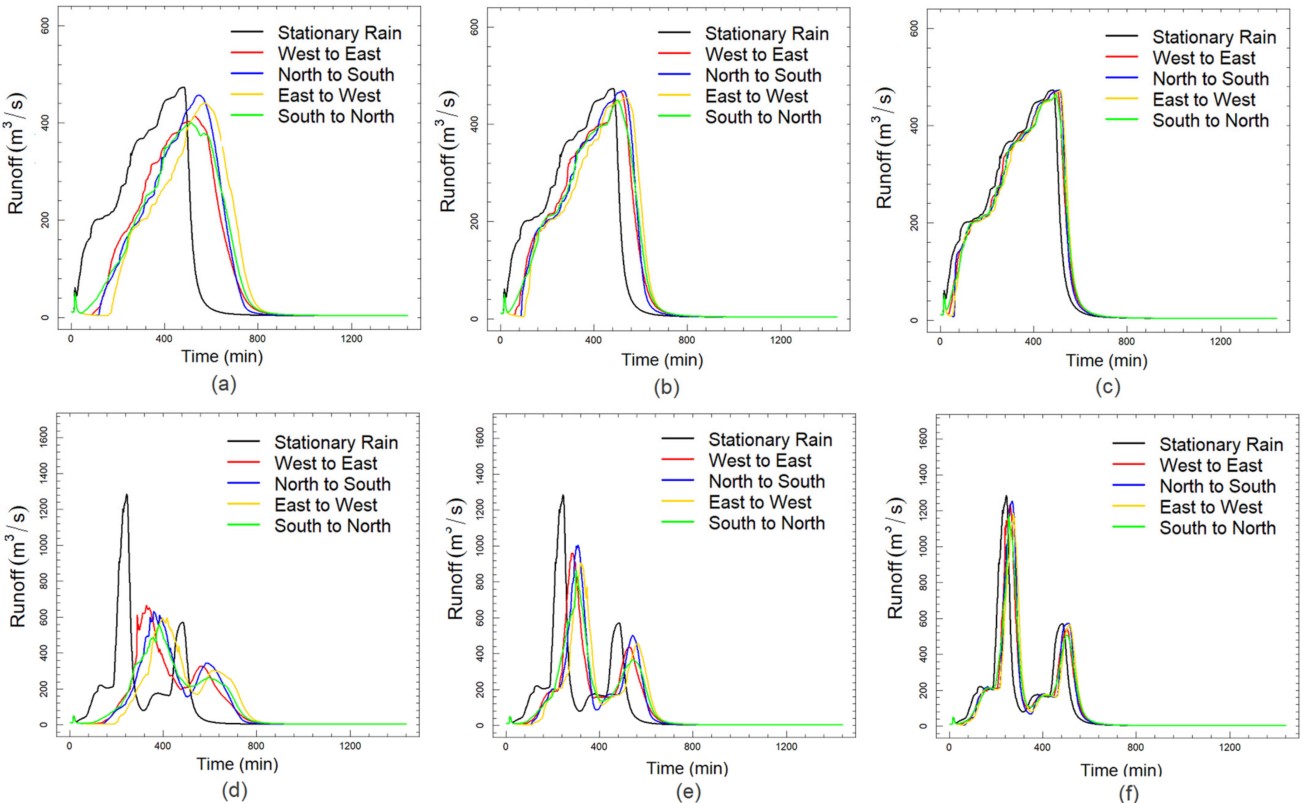

**Figure 12.** Hydrographs (at measuring point A) for rain variabilities $\sigma^2 = 0$ (**a**–**c**) & 99.2 (**d**–**f**) for stationary and non-stationary rain setups with velocities $v = 5$, 10, and 25 km/h (from left to right respectively) for storms moving in four directions.

The direction of rainfall movement affects the flooded areas in the two lower flood-plains, as shown in Figure 13 for $\sigma^2 = 0$ (solid line) and $\sigma^2 = 99.2$ (dashed line) rains under different velocities. Figure 13a–d show the absolute flooded areas and Figure 13e–i present the flooded areas normalized by the flooded areas of the stationary setup. Similar to Figure 11, faster storms tend to produce larger flooded areas in comparison to slower moving storms, regardless of directions. Different storm directions also seem to result in larger differences in flooded areas in presence of higher variated rainfall ($\sigma^2 = 99.2$) in comparison to lower variated rain ($\sigma^2 = 0$). This suggests that rain hyetograph variability has an amplifying effect on the role of storm direction in the resulting flooded areas. In both absolute and normalized figures, storms moving from north to south (the average main direction of the catchment stream) produce greater flooded areas in comparison to the other directions. Interestingly, when the storm is fast ($v = 25$ km/h) and storm movement is in the direction of the main stream (north to south), there are larger flooded areas in

comparison to stationary rainfall. This highlights the importance of considering storm movement while having fast moving storms in the direction of the main catchment stream.

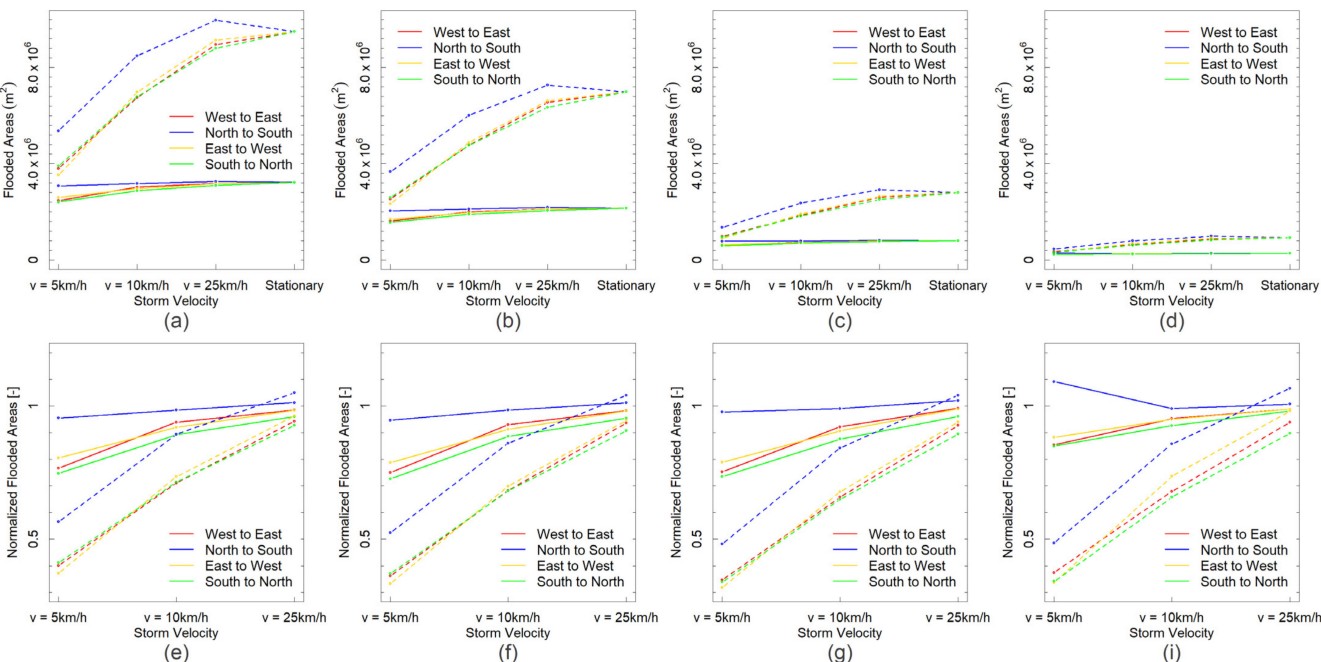

**Figure 13.** Flooded areas (**a–d**) and normalized flooded areas by the stationary setup (**e–i**) in the two lower floodplains for rain variabilities $\sigma^2 = 0$ (solid line) and $\sigma^2 = 100$ (dashed line) under the stationary and moving setups with velocities $v = 5$ km/h, $v = 10$ km/h, $v = 25$ km/h with four movement directions. Figure from left to right present water levels above 10 cm, 20 cm, 50 cm and 100 cm respectively.

For the sake of completeness, and to assess the robustness of these results to the original simplifying assumptions on infiltration and roughness, an additional set of simulations were run considering heterogeneous roughness and a uniform and constant infiltration rate, using the $\sigma^2 = 0$ & 99.2 rain signals. Simulations were run with stationary rain and storms moving from west to east with velocities of 5, 10, and 25 km/h. Runoff hydrographs under only roughness and infiltration plus roughness conditions are shown in Figure 14.

Heterogeneous roughness from the area is used in the simulations based on the land-use map offered by Ghorbanian et al. [77]. Manning values from Papaioannou et al. [10] have been applied. Infiltration is applied based on the Soil Conservation Service's (SCS) Curve Number (*CN*) method [83]. The cumulative rainfall is calculated for a certain hyetograph at each time interval using the following equation:

$$Q = \frac{(P - I_a)^2}{P - I_a + S} \tag{3}$$

In which $Q$ represents the cumulative excess rainfall, $P$ is the cumulative rainfall depth, $I_a = \min(P, 0.2S)$ is the initial abstraction, and $S$ represents the potential maximum retention which, if millimeters are used as units for rainfall depth, is calculated as:

$$S = 25.4\left(\frac{1000}{CN} - 10\right) \tag{4}$$

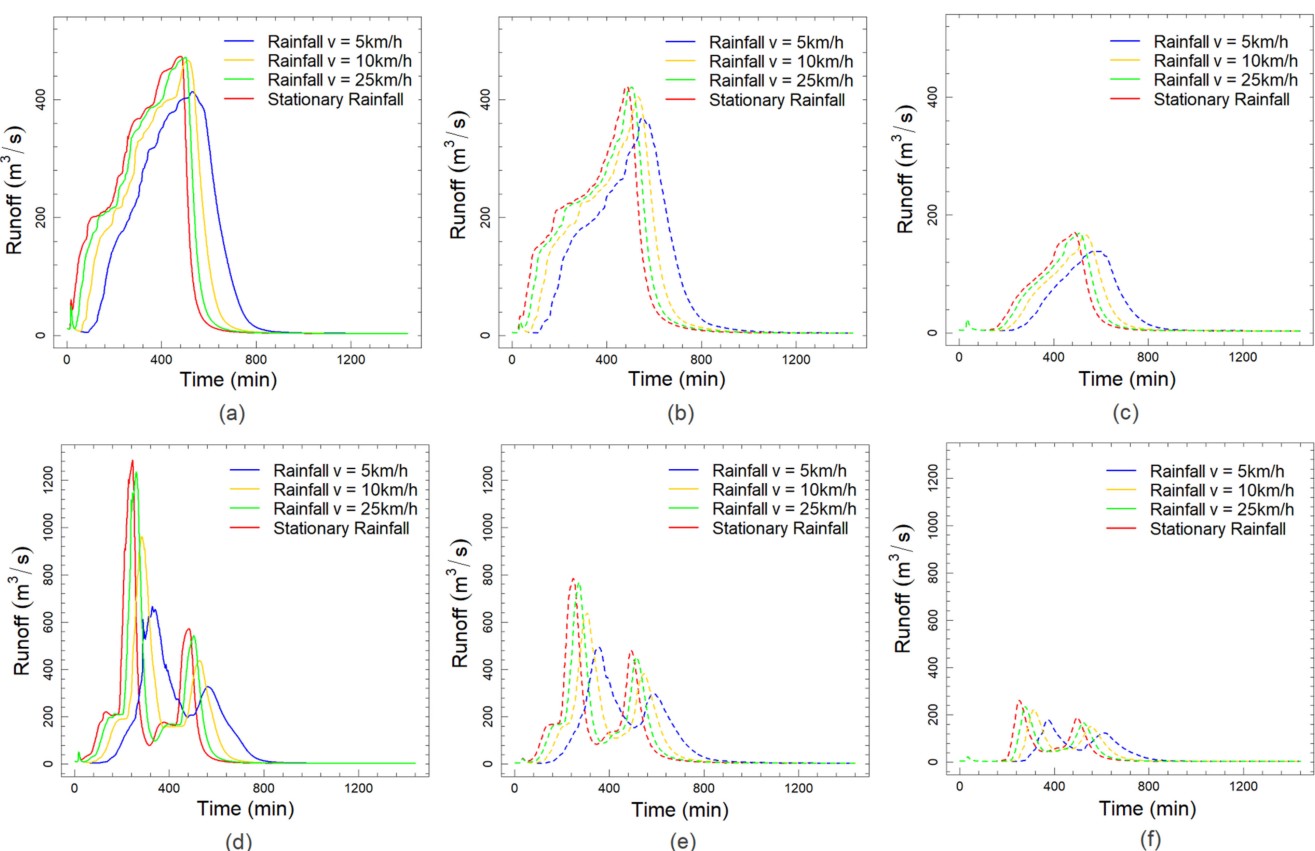

**Figure 14.** Hydrographs at measuring point A for rains $\sigma^2 = 0$ (Figure 3a) (**a–c**) and $\sigma^2 = 99.2$ (Figure 3f) (**d–f**) under an idealized setup (**a,d**), with consideration of heterogeneous roughness (**b,e**) and setup with consideration of both roughness and infiltration (**c,f**). Storm movement is from west to east with velocities $v = 5$, 10, and 25 km/h. Stationary rainfall scenarios are also included.

A homogeneous *CN* value of 81, which is suitable for arid and semiarid rangelands [84] was chosen for the upper catchment. While this may be a strong simplification of the infiltration process, since the objective of running these simulations is to compare to an impervious case, the next simplest representation of infiltration (uniform) suffices. Although it would be possible to implement more sophisticated infiltration models (e.g., Horton, Green-Ampt), this would open the parameter space, which is not the focus of this investigation. Previous studies have shown the value of taking such a stepwise approach to include additional processes, as it helps reveal how processes manifest in the hydrological signatures [42,60,74,85,86].

In general, a dampening effect is seen in the hydrographs as a result of infiltration and roughness, in the form of lower peaks and delayed runoff. Nonetheless, previous observations hold: the differences between stationary and moving rainfall are still present in both setups considering roughness, and roughness and infiltration. Namely, stationary rainfall achieves higher peak discharge and a faster onset of runoff compared to the moving storms. Faster storms still produce higher peaks and faster onsets and peak runoff compared to slower storms. Additionally, the rising limb of the damped hydrographs show two characteristic slopes, possibly related to the sequence of flow peaks collecting in the main streams, which is not evident in the impervious simulations.

*Limitations and Outlook*

One limitation of this study is the use of a single catchment as topography. While it is known that topography can influence runoff generation in many ways, such as affecting the onset of runoff [43], runoff connectivity [87], flow and transport pathways [88], drainage networks [89], etc., it has also been pointed out that the effects of rainfall spatiotemporal

variability are case dependent and may vary from catchment to catchment [36]. Therefore, exploring the effects of storm movement under a higher number of both natural and synthetic catchments may be worthwhile for reaching more generalized conclusions. This work is also consistent in offering an outlook towards further exploration of the interaction of storm movement with different catchment properties such as vegetation cover [90,91], climate [92], and microtopography [93] on runoff generation.

Another limitation of this study is the use of rainfalls with a single duration and volume which was chosen to enable a plausible comparison to be made among different scenarios. Although it is expected that the general effects of storm movement seen in this study will still be present under different rainfall durations and volumes, it is worthwhile exploring how different storm properties such as volume, duration, velocity, etc. can modulate this response.

Furthermore, this study focused on one definition of rainfall movement, which defined rainfall movement as storms moving at different speeds over the area, despite having the same rainfall duration and volume at every point of the watershed. This definition was chosen to increase comparability among different storm movement scenarios. Further exploring other definitions of storm movement with varying volumes and intensities based on the different properties of the movement may further advance our understanding.

## 4. Summary and Conclusions

The main objective of this work was to study how the use of moving hyetographs may affect hortonian runoff generation and, in particular, flash flood modeling results in comparison to the use of stationary rainfall. This was achieved through a theoretical modeling workflow by means of spatially distributed rainfall/runoff simulations. The Kan catchment in Iran was used as base topography for the simulations. Due to the catchments steep and highly contrasted topography and the upper regions of the catchment receiving high volumes of rain, the catchment shows high potential for allochthonous flash floods, which enabled studying the interactions between flash flood generation and rainfall movement through a modeling process. Fifty synthetic rain signals were generated using a random cascade model and traversed with different velocities and directions over the catchment area.

The results of this study indicate that storm movement can generate differences in the runoff generation process and in resulting flooded areas on the true topography of a real catchment. In particular, both the peak discharge and the early stages of runoff—the onset of runoff—are affected. Flooded areas are also shown to be sensitive to both the velocity and the direction of the storm movement. As expected, faster moving storms result in higher peaks and earlier onsets of runoff in comparison to slower moving storms. However, at a sub catchment level, mostly the onset was affected by storm velocity, suggesting that the sensitivity of peak discharge to storm movement may be dependent on the catchment scale. The direction of the storm movement was also shown to be a governing factor with storms moving in the direction of the main stream, producing the highest peaks and flooded areas. This is mainly in line with previous findings [26,28,31]. In addition to previous findings, rainfall hyetograph variability was also shown to interact with storm direction, amplifying its effect and importance under higher variated rains.

Additionally, the results confirm that storm movement and temporal variability can both affect flash flood response. Consequently, modeling efforts should account for these interactions. The flood response to rain signals with low temporal variability, exhibit less sensitivity to storm velocity and direction over the catchment, while producing less differences both in the hydrographs and flooded areas. However, the flood response to rain signals with high temporal variabilities show more sensitivity to storm movement properties. This clearly highlights the importance of accounting for rainfall movement in rainfall/runoff simulations and in particular in flash flood modeling while dealing with rainfall events of high temporal variability. Furthermore, it was shown that the sensitivity to storm movement and variability varies depending on the measurement location in the

catchment. Peak discharge was shown to be less sensitive to storm movement at sub catchment levels in comparison to the whole catchment. Additionally, floodplains and rivers in dry catchment areas may act as a filter and decrease the significance of rainfall temporal variability and storm movement in the generated hydrographs. This is a feature which may be particularly relevant in allochthonous rivers, making flood response and risk assessment potentially rather sensitive to the choice of discharge gauging locations.

Another indication from the results of this study is that, in many scenarios, stationary rainfall, which is typically used as design rainfall, often results in higher peaks and flooded areas. While using such a conservative approach may be classically preferred civil engineering [94], the cost efficiency of flood mitigation measures is also known to be an important factor to be taken into account in flood risk management [95,96]. Additionally, overdesign in structural flood protection can result in increased damage in the event of failure. It should be also noted that, as can be inferred from the results here, storm movement and variability interact with the spatial scales of the catchments. It is reasonable to expect that hydrographs are even more sensitive to storm movement and speed at larger catchment sizes. The results here also suggest that probabilistic flash-flood risk analysis should include, as best practice, the analysis of a plethora of storm designs considering temporal variability and storm movement.

While the presented study offers insight into the effects of storm movement on the resulting hydrographs and flooded areas in hydrodynamic modeling, given that every catchment system (and corresponding model) is unique [97], further studies concerning additional catchments and modeling approaches would be beneficial.

**Supplementary Materials:** The following supporting information can be downloaded at: https://www.mdpi.com/article/10.3390/w14121844/s1, Figure S1: Hydrographs (at measuring point A) for four generated rains (Figure 3a,b,e,f) under stationary and moving setups with velocities $v$ = 5 km/h, $v$ = 10 km/h, $v$ = 25 km/h with a direction from west to east under the storm definition provided by Ogden et al. (1995), in which the intensities of the storms are proportional to their velocity.

**Author Contributions:** Conceptualization, S.K.B.G. and D.C.-V.; methodology, S.K.B.G.; software, D.B. and S.K.B.G.; validation, S.K.B.G.; formal analysis, S.K.B.G. and D.C.-V.; investigation, S.K.B.G.; writing—original draft preparation, S.K.B.G.; writing—review and editing, S.K.B.G., D.B., D.C.-V. and C.H.; visualization, S.K.B.G.; supervision, D.B., D.C.-V. and C.H.; project administration, D.B.; funding acquisition, D.B. All authors have read and agreed to the published version of the manuscript.

**Funding:** This research was funded by Federal Ministry of Education and Research (BMBF) with grant number 13N15180 under the project HOWAMAN.

**Data Availability Statement:** Not applicable.

**Acknowledgments:** This research was supported by the German Federal Ministry of Education and Research (BMBF) in the "International disaster and risk management"-call (IKARIM) as part of the "Research for Civil Security" framework program under the project HOWAMAN.

**Conflicts of Interest:** The authors declare no conflict of interest.

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
