# Peer review of "Impact of Rainfall Movement on Flash Flood Response: A Synthetic Study of a Semi-Arid Mountainous Catchment"

_water, doi:10.3390/w14121844_

Round 1

Author Response

Dear reviewer,

Thank you for reading our manuscript and for your feedback and comments. Please find below our replies to your comments:

Regarding your comment on Figure 2, Thank you for pointing this out. We completely see your point, however due to other workflows in the paper such as the measuring points being mentioned in many figures and text, it was not feasible for us to change the naming during the short time for revision. However for the sake for clarity and also for making locating the points more comfortable for the readers, we have labelled the points in the map.

Regarding your comment on the long sentence in line 274, The sentence is now broken up in to two sentences. It is now located in line 282.

Regarding your comment for fitted graphs in Figure 4, Thank you for your comments, however it is seen that in figures b and c the spread of the data is relatively large. Although a trend is seen in the figures, the behavior doesn’t provide any useful information and is not of particular interest for this study. However an explanation on the relations between the three indicator and variance has been added in line 343.

Regarding your comment on Figure 5, The figure description is now updated to be more clear for the readers.

Regarding your comment on Figure 6 and the other figures related to placing the stationary rainfall before the moving rains in the figure legends, The main reason that we chose this order for the legends was to clarify that stationary rainfall is a different case than the moving rainfall and that the legend order does not necessarily show a progression in storm velocities.

Regarding your comment on line 476, The first point here is that Figure 9 is showing time of peak discharge for storms moving with a direction of west to east (dominant wind direction of the area) not from north to south. This is pointed out in the figure description. Additionally we added a short explanation why the peak discharge occurs faster compared to moving storm in line 500.

Regarding your comment on Figure 9, So the figures illustrate the time of peak discharge in the moving rainfall setups (with a movement direction from west to east) normalized by the time of peak in the stationary rainfall scenario. The normalized time of peak of the stationary rainfall (which after normalization has a value of 1) is plotted to make the comparison easier for the readers.

Regarding your comment on the size and readability of figure captions and text, Thank you for pointing this out. In most figures of the paper the text and labels are now increased in size.

Regarding your comment on Figure 11, The figure description is now updated to be more clear for the readers.

Regarding your comments on formatting and English corrections, Thank you so much for your detailed feedback, The changes have all been applied throughout the paper. The only exception is your comment on line 221 regarding changing 2D-2D to 1D-2D. In this case we are pointing out the coupling between the 2D floodplains in our model.

In Addition, the major made changes are:

  • The reference formatting is now changed to a number reference style throughout the text, which is required by the journal

Reviewer 2 Report

The authors explored the sensitivity of moving design storm hyetographs in the hydrologic response of a watershed governed by hortonian runoff generation and prone to occurrence of flash floods. Although the lack of observed data  imposes limitations, the authors designed adequate experiments and assumptions to overcome this issue. The paper includes a good review of the state of the art up to date. The results of the study are very interesting and include a good discussion. Giving these considerations, I think the paper is very suitable to be published by the journal. I have minor suggestions included below

Line 298 : measure by : described by

In general, increase the size of the labels in the figures 4, 7, 8, 9, 11, 12,13,14. Those are hard to read.

Line 644. Text in green

Author Response

Dear reviewer,

Thank you for reading our manuscript and for your feedback and comments. Please find below our replies:

“Line 298 : measure by : described by”, the word is now replaced in line 308.

“In general, increase the size of the labels in the figures 4, 7, 8, 9, 11, 12,13,14. Those are hard to read”, Thank you for pointing this out. The label and texts in all figure have become bigger now.

“Line 644. Text in green”, the coloring is now corrected.

In Addition, the major changes made are:

  • The reference formatting is now changed to a number reference style throughout the text, which is required by the journal
  • In Figure 2, the measuring point labels are now written in the figure for the readers comfort
  • Some small English and formatting corrections throughout the text 

Reviewer 3 Report

This paper is within the scope of the Water journal; I suggest a major revision.

Comments:

  1. Many previous studies related to the flash floods forecast in the mountain and urban should be considered in the Introduction section, e.g., Tzu-Yin Chang, Hongey Chen, Huei-Shuin Fu, Chen, Wei-Bo Yi-Chiang Yu,Wen-Ray Su, Lee-Yaw Lin. 2021. An Operational High-Performance Forecasting System for City-Scale Pluvial Flash Floods in the Southwestern Plain Areas of Taiwan. Water, 13(4); Yung-Ming Chen, Che-Hsin Liu, Hung-Ju Shih, Chih-Hsin Chang, Chen Wei-Bo, Yi-Chiang Yu, Wen-Ray Su, Lee-Yaw Lin, 2019. An Operational Forecasting System for Flash Floods in Mountainous Areas in Taiwan. Water, 11(10), 2100. (SCIE).
  2. The authors have to explain the connections between 1D and 2D models and their validations.

Author Response

Dear reviewer,

Thank you for reading our manuscript and for your feedback and comments. Please find below our replies:

“Many previous studies related to the flash floods forecast in the mountain and urban should be considered in the Introduction section, e.g., Tzu-Yin Chang, Hongey Chen, Huei-Shuin Fu, Chen, Wei-Bo Yi-Chiang Yu,Wen-Ray Su, Lee-Yaw Lin. 2021. An Operational High-Performance Forecasting System for City-Scale Pluvial Flash Floods in the Southwestern Plain Areas of Taiwan. Water, 13(4); Yung-Ming Chen, Che-Hsin Liu, Hung-Ju Shih, Chih-Hsin Chang, Chen Wei-Bo, Yi-Chiang Yu, Wen-Ray Su, Lee-Yaw Lin, 2019. An Operational Forecasting System for Flash Floods in Mountainous Areas in Taiwan. Water, 11(10), 2100. (SCIE).”, The mentioned papers are now cited and included in the introduction of the paper at line 53.

“The authors have to explain the connections between 1D and 2D models and their validations.”, The connection between the 1D and 2D models have been explained more in detail in lines 222-227

In Addition, the major changes made are:

  • The reference formatting is now changed to a number reference style throughout the text, which is required by the journal
  • The labels and text in most figures have seen an increase in size for the readers comfort
  • In Figure 2, the measuring point labels are now written in the figure for the readers comfort
  • Some small English and formatting corrections throughout the text 

Round 2

Reviewer 3 Report

The authors have well addressed my comments and concerns about this manuscript; additionally, the authors significantly improve the manuscript. I, therefore, suggest that this manuscript can be accepted for publication in the Water journal in the present form.